# Proton Migration on Top of Charged Membranes

**DOI:** 10.3390/biom13020352

**Published:** 2023-02-11

**Authors:** Ewald Weichselbaum, Timur Galimzyanov, Oleg V. Batishchev, Sergey A. Akimov, Peter Pohl

**Affiliations:** 1Institute of Biophysics, Johannes Kepler University Linz, 4040 Linz, Austria; 2A.N. Frumkin Institute of Physical Chemistry and Electrochemistry, Russian Academy of Sciences, Moscow 119071, Russia; 3Department of Theoretical Physics and Quantum Technologies, National University of Science and Technology “MISiS”, Moscow 119991, Russia

**Keywords:** bioenergetics, local proton circuit, proton transport, fluorimetry, caged proton, interfacial proton diffusion

## Abstract

Proton relay between interfacial water molecules allows rapid two-dimensional diffusion. An energy barrier, $\Delta G_{r}^{‡}$, opposes proton-surface-to-bulk release. The $\Delta G_{r}^{‡}$-regulating mechanism thus far has remained unknown. Here, we explored the effect interfacial charges have on $\Delta G_{r}^{‡}$’s enthalpic and entropic constituents, $\Delta G_{H}^{‡}$ and $\Delta G_{S}^{‡}$, respectively. A light flash illuminating a micrometer-sized membrane patch of a free-standing planar lipid bilayer released protons from an adsorbed hydrophobic caged compound. A lipid-anchored pH-sensitive dye reported protons’ arrival at a distant membrane patch. Introducing net-negative charges to the bilayer doubled $\Delta G_{H}^{‡}$, while positive net charges decreased $\Delta G_{H}^{‡}$. The accompanying variations in $\Delta G_{S}^{‡}$ compensated for the $\Delta G_{H}^{‡}$ modifications so that $\Delta G_{r}^{‡}$ was nearly constant. The increase in the entropic component of the barrier is most likely due to the lower number and strength of hydrogen bonds known to be formed by positively charged residues as compared to negatively charged moieties. The resulting high $\Delta G_{r}^{‡}$ ensured interfacial proton diffusion for all measured membranes. The observation indicates that the variation in membrane surface charge alone is a poor regulator of proton traffic along the membrane surface.

## 1. Introduction

Interfacial proton diffusion is a widespread phenomenon. It is known as spillover in catalysis, where protons are created on a metal surface and migrate to the surface of the support [1,2]. Proton migration has also been observed on top of biological membranes [3,4], artificial membranes [5,6,7], hydrophobic interfaces [8], proteins [9], graphene sheets [10], hexagonal boron nitride crystals [11], hydrophilic alumina and silica surfaces [12], and silicon nitride [13]. Most strikingly, the laterally diffusing protons do not freely exchange with bulk protons [14].

Judging from the long distances, *s*, protons may travel along lipid bilayers, and the energy barrier $\Delta G_{r}^{‡}$ opposing proton release into the bulk may be substantial. The time, *t*_d_, it takes the proton to cross large distances allows estimating $\Delta G_{r}^{‡}$. For example, the observed surface diffusion coefficient, *D* = 5.8 × 10^−5^ cm^2^ s^−1^ [5], allows finding *t*_d_ with the help of Equation (1):(1)$$t_{d}=\frac{s^{2}}{4D}$$

For the reported value *s* = 125 µm [5], we calculate *t*_d_ ≈ 0.7 s. Employing transition state theory, we may find $\Delta G_{r}^{‡}$ as follows:(2)$$\Delta G_{r}^{‡}=-k_{B}T\ln\left(\frac{k_{\mathrm{off}}}{\nu_{0}}\right)$$
where *T*, *k*_B_, and $\nu_{0}$ ≈ 10^13^ s^−1^ are the absolute temperature, the Boltzmann constant, and the universal transition state theory attempt frequency for rate processes at surfaces [15], respectively. Assuming that the rate, k_off_, of proton surface-to-bulk release is equal to 1/*t*_d_ = 1.4 s^−1^, Equation (1) returns $\Delta G_{r}^{‡}$ ≈ 30 *k*_B_*T*.

Such high $\Delta G_{r}^{‡}$ is surprising, as it is comparable to the apparent activation energies Δ*E*_A_ for the membrane permeation of small uncharged molecules, e.g., water [16,17]. Compared to the much higher membrane barrier for cations, protons face an unusually small Δ*E*_A_ [17] of only 22–37 *k*_B_*T* for membrane transfer [18,19,20]. However, Δ*E*_A_ and $\Delta G_{r}^{‡}$ are still of similar size. Importantly, Δ*E*_A_ describes a reversible thermodynamic process, whereas proton surface-to-bulk release is irreversible [21]. Accordingly, the quantitative agreement of an equilibrium description [6] for the proton release process is poor compared to a non-equilibrium model [22]. An attempt to use equilibrium power-law desorption kinetics [23] to model proton interfacial diffusion as a repetition of multiple desorption and re-adsorption events also failed [21].

The non-equilibrium model adequately reflects the change of proton concentration, *σ,* in the water layers adjacent to the membrane at time *t* [22]:(3)$$\sigma \left(x,t\right)=\sigma_{0}+\frac{A_{n}}{4\pi Dt}\exp\left(-\frac{x^{2}}{4Dt}\right)\exp\left(-t\;k_{\mathrm{off}}\right)\;$$
where *D* is the interfacial (lateral, two-dimensional) proton diffusion coefficient, and *k*_off_ is the release rate coefficient from the membrane surface. *A*_n_ stands for the proton concentration increase, and *σ*_0_ denotes the pre-existing proton concentration adjacent to the surface.

Importantly, $\Delta G_{r}^{‡}$ does not originate from proton binding to titratable residues on the membrane surface [6]. Neither removal of the titratable residues [6] nor replacement of the membrane with an organic solvent [8] abolishes interfacial proton diffusion. Monitoring the uptake and release kinetics of titratable residues on the membrane surface [24] or solid surfaces [25] may yield interfacial proton diffusion coefficients that are orders of magnitude too small [6]. The underestimation is because proton hopping between titratable residues reflects the time required for proton release, which is in the order of 1 ms for residues with neutral pK_a_ values [26]. Yet, a proton’s residence time on an interfacial water molecule visited during interfacial proton hopping is much shorter. Assuming an average distance between water molecules of ~0.3 nm and the above-mentioned surface diffusion coefficient of *D* = 5.8 × 10^−5^ cm^2^ s^−1^ [5], Equation (1) yields *t*_d_ = 0.4 ns.

In line with the above considerations that proton hopping occurs between water molecules [5,6,8], $\Delta G_{r}^{‡}$ contains only a minor enthalpic constituent $\Delta G_{H}^{‡}$. It is in the order of only one hydrogen bond [21], thereby ensuring high lateral proton mobility. Besides being attributed to structured water [8], the exact molecular origin of the much larger entropic constituent, $\Delta G_{s}^{‡}$, barrier thus far remained elusive. Ab initio dynamics simulations suggested that the preferable 150° orientation of the interfacial water dipoles with respect to the interface normal contributes to $\Delta G_{s}^{‡}$ [8,27], as the interactions of the surface proton with the electron lone pairs of distinctly oriented interfacial water molecules may play a key role for proton transport.

Measurements of the so-called membrane dipole potential have long indicated a preferential water orientation at the lipid bilayer interface. Oriented interfacial water molecules contribute roughly 100 mV to the membrane dipole potential [28,29,30]. Interestingly, the contribution appears not to depend on the lipid charge as if phosphatidyl moieties or negatively charged headgroups were not forcing water reorientation.

In contrast, molecular dynamics simulations show variances in the organization of water–lipid interfaces in dioleoylphosphatidylserine (DOPS) and dioleoylphosphatidylcholine (DOPC) bilayers [31]. Thanks to the looser-packed water–lipid interface of the DOPC bilayer, water molecules penetrate deeper inside the zwitterionic (DOPC) bilayer, enabling their interaction with lipid carbonyls. Infrared-visible sum frequency spectroscopy also revealed structural differences between interfacial water near neutral and negatively charged supported lipid bilayers [32]. Raman solvation shell spectroscopy and second harmonic scattering (SHS) show that the hydration shells and the interfacial water structure are very different for positively and negatively charged amphiphilic ions [33].

These diverging observations suggest that membrane surface charges may act twofold on proton surface diffusion: They may (i) electrostatically interact with the surface proton, thereby altering $\Delta G_{H}^{‡}$_,_ and (ii) alter the structure of the interfacial water in a way that affects $\Delta G_{s}^{‡}$. Here, we tested both hypotheses using free-standing planar lipid bilayers. Upon illumination with UV light, a hydrophobic caged proton released the protons in a local membrane patch. We monitored their diffusion to a distant membrane patch as a function of membrane composition and temperature. The surface charge affected $\Delta G_{H}^{‡}$, as may be predicted from Gouy-Chapman’s diffuse double-layer theory. Since $\Delta G_{s}^{‡}$ exhibited compensatory changes, the remaining energy barrier enabled surface proton diffusion even in the case of a proton-repelling, positively charged membrane.

## 2. Materials and Methods

The experimental setup has been described previously [5,6]. In brief, we formed horizontal planar lipid bilayers from a solution of 20 mg lipid solution in 1 mL *n*-decane (Sigma-Aldrich, Vienna, Austria) in a 200–300 µm wide aperture of a Teflon septum. For negatively charged membranes, we used 1,2-dioleoyl-sn-glycero-3-phosphoglycerol (DOPG, Avanti Polar Lipids, AL, USA); for positively charged membranes, we exploited a 1:1 mixture of 1,2-dioleoyl-3-trimethylammonium-propane (DOTAP, Avanti Polar Lipids) and 1,2-dioleoyl-sn-glycero-3-phosphocholine (DOPC, Avanti Polar Lipids). The membrane-forming solution contained ~1 mol % of the pH-sensor fluorescein covalently linked to N-(Fluorescein-5-thiocarbamoyl)-1,2-dihexadecanoyl-sn-glycero-3-phosphoethanolamine (Fluorescein DHPE, ThermoFisher, Waltham, MA, USA) and a caged-proton compound, (6,7-dimethoxycoumarin-4-yl)methyl diethyl phosphate [34]. A UV light pulse (<400 nm) emitted by a xenon flash lamp (Rapp OptoElectronic; Wedel; Germany) released the protons. After passing a diaphragm (Till Photonics, Gräfelfing, Germany), it illuminated a 10 × 10 µm^2^ membrane patch. Light from a second xenon lamp (150 W) went through a monochromator (put to 488 nm) and then through another diaphragm. It excited the Fluorescein DHPE FPE fluorescence in a 10 × 10 µm^2^ large membrane area. The emitted light first passed a 515 nm high-pass filter and then hit a photomultiplier (Till Photonics). The buffer consisted of 10 mM KCl and 0.1 mM Capso (3-(Cyclohexylamino)-2-hydroxy-1-propanesulfonic acid) or 0.065 mM Tricine (N-(Tri(hydroxymethyl)methyl)glycin). pH was adjusted to 9.0 or 8.0, respectively.

### Determination of Membrane Surface Potential

We produced large unilamellar vesicles as described previously [35]. In brief, we rehydrated a flask-covering lipid film with the same buffer solution we used in the planar bilayer experiments. Extrusion (19 times) of the resulting suspension through 100 nm filters produced large unilamellar vesicles. Using a Delsa Nano Particle Analyzer (Beckman Coulter, Krefeld, Germany), we subsequently determined the ζ potential via measurements of electrophoretic vesicle mobility [36]. To calculate the *ψ*_0_ from ζ, we used the Mathematica (Wolfram, Champaign, IL, USA) routine NSolve to find the solution for x = 0 of the following equation [37]:(4)$$\;\psi \left(x\right)=\frac{2k_{B}T}{ze}ln\left(\frac{1+\alpha \exp\left(-\kappa x\right)}{1-\alpha \exp\left(-\kappa x\right)}\right)$$

Here, *ψ*(*x* = 0.4 nm) = ζ. 1/*κ* is defined as the Debye length.

## 3. Results

We painted a horizontal planar lipid bilayer in the aperture of a Teflon septum and allowed a caged-proton compound, (6,7-dimethoxycoumarin-4-yl)methyl diethyl phosphate, to adsorb to it [34]. Exposing a 10 × 10 µm^2^-sized membrane area to a UV flash (Rapp OptoElectronics, Berlin, Germany) produced a proton wave [5]. The decrease in fluorescence intensity of lipid-anchored fluorescein indicated H^+^ arrival at a distant 10 × 10 µm^2^-sized spot. As the protons moved on, the fluorescence intensity increased again. The time elapsing between protons release and the observation of a fluorescence minimum depended on (i) the distance *x* of proton travel and (ii) the interfacial proton diffusion coefficient *D*, also called lateral diffusion coefficient (Figure 1).

We noticed a slightly delayed arrival of the proton wave on charged membranes compared to the zwitterionic DOPC bilayer (Figure 1). Relative to DOPC membranes, the amplitude of the arriving proton wave increased adjacent to DOPG membranes and decreased close to DOTAP/DOPC membranes (Figure 1). In the former case, the electrostatic attraction by the negatively charged lipids was responsible. In the latter case, the electrostatic repulsion by the positively charged membrane increased interfacial pH, resulting in a much-reduced sensitivity of the membrane-anchored pH sensor, fluorescein. This observation prompted us to decrease bulk pH to pH 8 in the subsequent experiments with the DOTAP/DOPC mixture.

To determine *D*, we first converted fluorescence intensity into the proton concentration [H^+^] of the membrane-adjacent water layers with the help of calibration curves that depicted the fluorescence intensity (normalized to peak fluorescence intensity) as a function of bulk pH. We measured these curves for the individual lipid compositions with a vesicle suspension in equilibrium. The subsequent calculation of proton concentration accounts for membrane surface potential $\psi_{0}$ and thus for the steady-state proton concentration difference between the bulk and the bilayer surface. Second, we normalized [H^+^] using the maximal values; third, smoothed the experimental data employing an exponential moving average; fourth, we resampled the data down, taking only 300 points out of 50,000 original ones; and fifth, we fitted a modified version of Equation (1) to the thus-attained data. Equation (1)’s modification accounts for the finite size of the proton release and detection areas:(5)$$\sigma_{a}\left(x^{\prime },y^{\prime },t\right)=\sigma_{a,0}+\frac{\sigma_{sq}}{4}\left(\mathrm{erf}\left(\frac{x^{\prime }+h}{2\sqrt{Dt}}\right)-\mathrm{erf}\left(\frac{x^{\prime }-h}{2\sqrt{Dt}}\right)\right)\left(\mathrm{erf}\left(\frac{y^{\prime }+h}{2\sqrt{Dt}}\right)-\mathrm{erf}\left(\frac{y^{\prime }-h}{2\sqrt{Dt}}\right)\right)\exp\left(-k_{\mathrm{off}}t\right)$$
where *σ*_sq_ and 2*h* (=10 µm in our setup) are the relative [H^+^] increment right after the excitation and the side length of the excitation square, respectively. erf(*z*) is the Gauss error function:(6)$$\mathrm{erf}\left(z\right)=\frac{2}{\sqrt{\pi }}\int_{0}^{z}e^{-\varsigma^{2}}d\varsigma$$

We integrated Equation (5) within the boundaries [*x − h*, *x + h*] and [*−h*, *h*] for the variables *x′* and *y′*, respectively, since the fluorescent signal is gathered from a square region of the size 2*h* × 2*h*:(7)$$\sigma \left(x,t\right)=\frac{1}{4h^{2}}\int_{\left(x-h\right)}^{\left(x+h\right)}\int_{-h}^{h}\sigma_{a}\left(x^{\prime },y^{\prime },t\right)dx^{\prime }dy^{\prime }$$

Finally, Equation (5) retains the following analytical form:(8)$$\begin{array}{l} \sigma \left(x,t\right)=\sigma_{0}+A_{n}\exp\left(-k_{\mathrm{off}}t\right)\left[\mathrm{erf}\left(\frac{h}{\sqrt{Dt}}\right)h-\sqrt{\frac{Dt}{\pi }}\left(1-\exp\left(-\frac{h^{2}}{Dt}\right)\right)\right]\\ \times [\left(x-2h\right)\mathrm{erf}\left(\frac{x-2h}{2\sqrt{Dt}}\right)+\left(x+2h\right)\mathrm{erf}\left(\frac{x+2h}{2\sqrt{Dt}}\right)-2x\mathrm{erf}\left(\frac{x}{2\sqrt{Dt}}\right)\\ +2\sqrt{\frac{Dt}{\pi }}\left(\exp\left(-\frac{{\left(x+2h\right)}^{2}}{4Dt}\right)+\exp\left(-\frac{{\left(x-2h\right)}^{2}}{4Dt}\right)-2\exp\left(-\frac{x^{2}}{4Dt}\right)\right)] \end{array}$$

When globally fitting Equation (8) to the data, we assumed *D* and *k*_off_ to be equal for each dataset measured at similar temperatures and lipid compositions. Every dataset consisted of several measurements taken at different *x*-values, i.e., comprised of at least 16 independent records. Every record represented an average of ~20 sweeps. Figure 2 shows four representative fluorescent records (only one per *x*-value) for DOPG. Once the global fit returned *D* and *k*_off_ values with satisfactory accuracy, we moved to the next temperature, acquired new traces at different distances, and performed a new global fit.

We repeated the experiment in Figure 2 at different temperatures (Figure 3A). We performed a global fit for each temperature, including all traces obtained from distance-dependent measurements. An Arrhenius plot (Figure 3B), namely
(9)$$D=A_{D}e^{-\frac{\Delta H_{D}^{‡}}{k_{B}T}}$$
permitted calculation of the activation enthalpy, $\Delta H_{D}^{‡}$ = (4.5 ± 1.2) *k*_B_*T*. *A*_D_ is a pre-exponential coefficient. Within the error, $\Delta H_{D}^{‡}$ is equal to (i) the corresponding value obtained for DOPC membranes (5.9 ± 1.1) *k*_B_*T* [21] and (ii) the experimental activation enthalpy of 4.3 *k*_B_*T* for bulk proton mobility [38].

The Arrhenius plot for the proton release coefficient $k_{\mathrm{off}}$, namely
(10)$$k_{\mathrm{off}}=A_{k}e^{-\frac{\Delta H_{k}^{‡}}{k_{B}T}}\;$$
for DOPG membranes indicates $\Delta H_{k}^{‡}$ = (13 ± 3) *k*_B_*T* (Figure 3C). *A*_k_ is a pre-exponential coefficient. At first glance, the large value may appear surprising. Since the proton is first released and only subsequently chooses its direction of travel perpendicular or parallel to the membrane plane, $\Delta H_{k}^{‡}$ = $\Delta H_{D}^{‡}$ may have been expected. Yet, only the proton moving away from the membrane has to overcome the attractive electrostatic force exerted by the negative surface potential. Accordingly, we hypothesized that (i) $\Delta H_{k}^{‡}$ is a superposition of two terms: the activation enthalpy $\Delta H_{r}^{‡}$ for membranes with overall zero surface charge and the energy *E* of the electrical field generated by the surface charges:(11)$$\Delta H_{k}^{‡}=\Delta H_{r}^{‡}+E\;$$
where
(12)$$\Delta H_{r}^{‡}\approx \Delta H_{D}^{‡}$$
and
(13)$$E=-e\psi_{0}\;$$$\psi_{0}\;$ is the membrane surface potential, and *e* is the elementary charge. We calculated $\psi_{0}\;$ from the measured electrophoretic vesicle mobility of large unilamellar vesicles [36,37] of the same composition as our planar lipid bilayers using identical buffer conditions.

We found $\psi_{0}\;$ = –110 mV at 24 °C. Equation (13) returned *E* ≈ (4.3 ± 0.2) *k*_B_*T* (Table 1). Since $\Delta H_{r}^{‡}=\left(8.8\;\pm \;3.2\right)k_{B}T\;$ and $\Delta H_{D}^{‡}=\left(4.5\;\pm \;1.2\right)k_{B}T$ (Table 2), Equation (11) holds for DOPG bilayers; i.e., $\Delta H_{r}^{‡}$ is not significantly different from $\Delta H_{D}^{‡}$. As previously reported, the same result is also valid for the zwitterionic DOPC membranes [21].

The pre-exponential factors *A*_D_ and *A*_k_ allow assessment of the entropy of activation for proton release $\Delta S_{r}^{‡}$. From transition state theory, we anticipate the following [21]:(14)$$A_{k}=\nu_{0}\exp\left(\frac{\Delta S_{r}^{‡}}{k_{B}}\right)\;$$*ν*_0_ can be assessed using the Einstein relation (in two dimensions) $\nu_{0}=A_{D}\frac{4}{l^{2}}$, where *l* = 2.8 Å is the O-O distance in liquid water across which the proton hops. Therefore, the activation entropy for proton release can be estimated by the following relationship:(15)$$\Delta G_{s}^{‡}=T\Delta S_{r}^{‡}=k_{B}T\ln\left(\frac{A_{k}}{\nu_{0}}\right)=k_{B}T\ln\left(\frac{l^{2}A_{k}}{A_{D}4}\right)\;$$

For the DOPG system, we obtain $T\Delta S_{r}^{‡}=\left(-20\pm 3\right)k_{B}T$.

Encouraged by these results, we proceeded with measurements of positively charged membranes. Pure DOTAP planar membranes appeared to be relatively unstable and somewhat leaky, so we mixed the positively charged lipid with 50 mol % DOPC. As with pure DOPC and DOPG membranes, we averaged 20 subsequent proton release events to obtain one record. We obtained at least eight such recordings, each at 30 µm and 50 µm. After converting the fluorescence intensities into proton concentrations as described above, we globally fitted these records to extract *D* and *k*_off_ using Equation (8). In the next step, we repeated the same procedure for the three remaining temperatures. Figure 4A shows typical records of the proton concentration adjacent to the DOTAP:DOPC 1:1 membrane surface as well as global fits to the experimental data. An Arrhenius plot (Figure 4B) yielded $\Delta H_{D}^{‡}$ = 6.6 ± 0.9 *k*_B_*T*.

Unfortunately, the DOTAP/DOPC data did not show a temperature dependence of the release coefficient; i.e., all four global fits returned *k*_off_ = (1.2 ± 1.2) s^–1^ (Figure 4C). To compute $\Delta H_{r}^{‡}$, we measured the electrophoretic mobility of DOTAP/DOPC vesicles and calculated $\psi_{0}$ = +(90.5 ± 5.8) mV (Table 1). Accordingly, *E* amounts to (–3.5 ± 0.2) *k*_B_*T* (Table 1), decreasing the release activation barrier by the same value. Again, we can confirm the following:(16)$$\Delta H_{r}^{‡}=\Delta H_{k}^{‡}-E=\left(3.5\;\pm \;3.2\right)k_{B}T\;\approx \Delta H_{D}^{‡}=\left(6.6\;\pm \;0.9\right)k_{B}T$$

In other words, $\Delta H_{r}^{‡}$ is not significantly different from $\Delta H_{D}^{‡}$ for DOTAP membranes (Table 2).

## 4. Discussion

Our study questions the role of interfacial electrostatics as a regulator for proton migration along the membrane–water interface between a proton source and a proton sink. We show that the first step leading to proton release is independent of interfacial charges. It requires the breakage of only a single hydrogen bond. The corresponding energetic expense is reflected by the enthalpic components Δ*H*_D_ for lateral diffusion and Δ*H*_r_ for proton surface-to-bulk release. In lateral proton diffusion, the movement to a neighboring interfacial water molecule occurring in the second step is not associated with a major energetic expense. In contrast, the proton heading for the bulk solution invests or gains energy *E* when moving away from the bilayer in its second step since this movement is either opposed by attractive forces originating from negative charges or facilitated by repulsive forces due to the presence of positive charges. $\Delta H_{r}^{‡}$ and *E* sum up to $\Delta H_{k}^{‡}$.

We find that zeroing $\Delta H_{k}^{‡}$ is insufficient to obliterate lateral proton diffusion. Even for positively charged membranes, where $\Delta H_{k}^{‡}$ approaches zero, a significant energetic barrier opposes proton surface to bulk release. Judging from *k*_off_’s mean value of ~1 s^−1^, the barrier $\Delta G_{r}^{‡}$ is in the order of 30 *k*_B_*T* (compare Equation (2)).

Accordingly, $\Delta G_{r}^{‡}$ is very close to the value reported for zwitterionic membranes [21].

At first glance, this result contradicts the observation of a much shorter travel distance on top of DOPC/DOTAP membranes than pure DOPC membranes. However, the pH shift adjacent to the proton release site also decreased, indicating a decrement in caged proton concentration on top of the planar bilayer. This result is in line with more restricted hydration next to DOTAP membranes—a previously reported hypothesis based on DOTAP-induced pK_a_ shifts of membrane-adsorbed weak acids [39].

The decreased hydration level may also explain the compensatory increase of $\Delta S_{r}^{‡}$ in the case of positively charged membranes. The high $\Delta G_{r}^{‡}$ value derived according to Equation (2) indicates an increase of the barrier’s entropic component $T\Delta S_{r}^{‡}$ counteracting the decrease in $\Delta H_{r}^{‡}$. DOTAP substitutes 50% of the zwitterionic DOPC; i.e., it removes 50% of the more strongly hydrated phosphate moieties while leaving the density of the less-hydrated positively charged residues unaltered. Physiological consequences of the different hydration strengths between negatively and positively charged organic ions has previously been noted [40]. The observation that water forms far fewer and weaker bonds with positively charged ions than with negatively charged ions [41] was used, for example, to explain differences in aquaporin unitary water channel conductivities measured for channels with differently charged vestibules [42]. Yet, the hydration effect has not been previously associated with the height of the entropic barrier opposing proton surface-to-bulk release. The most probable reason is the relatively small increment of $\Delta G_{s}^{‡}$ and thus the comparatively small perturbation of the interface’s water-ordering effect. More considerable changes, i.e., obliterating ordered water, may inhibit interfacial proton travel, as water orientation appears necessary for lateral proton migration [8,27].

Our experiment with DOPG membranes confirms the above conclusion. The removal of all positively charged moieties, i.e., of choline headgroups and trimethylammonium headgroups, reduces $\Delta G_{s}^{‡}$ to 2/3 of its initial value. Yet, the remaining $\Delta G_{s}^{‡}$ is still about twice as large as the enthalpic component.

The considerable height of $\Delta G_{s}^{‡}$ ensures energy-efficient travel between a proton source and a proton sink on the same membrane. Such a local proton circuit is deemed especially important in bioenergetics [43,44]. For example, the low bulk electrochemical proton gradient would not efficiently drive the ATP synthase (proton sink) of alkalophilic bacteria [45]. For the calculation, it is sufficient to consider a cytoplasmic pH over two units below the high external pH and a membrane potential smaller than 200 mV [46]. Local proton circuits may also be essential for protein translocation facilitated by the SecY translocase [47]. Protein translocation is rather inefficient if driven solely by ATP hydrolysis instead of being accelerated by the electrochemical proton gradient [48]. The question is how the interfacial proton from the top of the membrane enters the proteinaceous proton sink. In cytochrome C oxidase, efficient proton transfer from the membrane surface to the K proton pathway may occur between the lipid phosphate group and the nearby glutamate 101 [49]. Substituting the acidic glutamate for an alanine decreased the proton transfer rate by a factor of five. The proton channel HV1 provides another example. Substitution of the acidic aspartate residue at position 174 for alanine reduced the transport rate of this channel by 2 ½ fold [50]. It is not entirely clear whether the acidic residues at the entryway into the conducting pores act as proton relay sites or merely increase the local proton concentration by electrostatic attraction. The low impact of acidic residues for interfacial proton travel revealed in our experiments (Figure 1, Figure 2 and Figure 3) would suggest the latter scenario. Yet, the emphasis on phosphate groups as relay sites for proton transfer made by molecular dynamics simulations [51,52] may favor the alternative idea of proton relay from phosphate to an acidic amino acid side chain. More work is required to obtain inside into the molecular mechanism of proton transfer between the membrane and a proteinaceous sink.

## 5. Conclusions

We conclude that cells relying on proton surface migration for efficient ATP production or proton-coupled transport cannot use charge-related hydration effects for fine-tuning proton membrane affinity. Moreover, dedicated membrane proteins expressed primarily to regulate interfacial proton travel have not yet been described. The role of proteins seems to be restricted to a short-term regulation of interfacial proton abundance, as has been shown for uncoupling proteins [53]. Thus, uncharged lipid-anchored moieties capable of strongly altering interfacial hydration may represent an energy-efficient solution. Further studies are on the way to identify such lipid regulators of proton traffic.

## Figures and Tables

**Figure 1 biomolecules-13-00352-f001:**
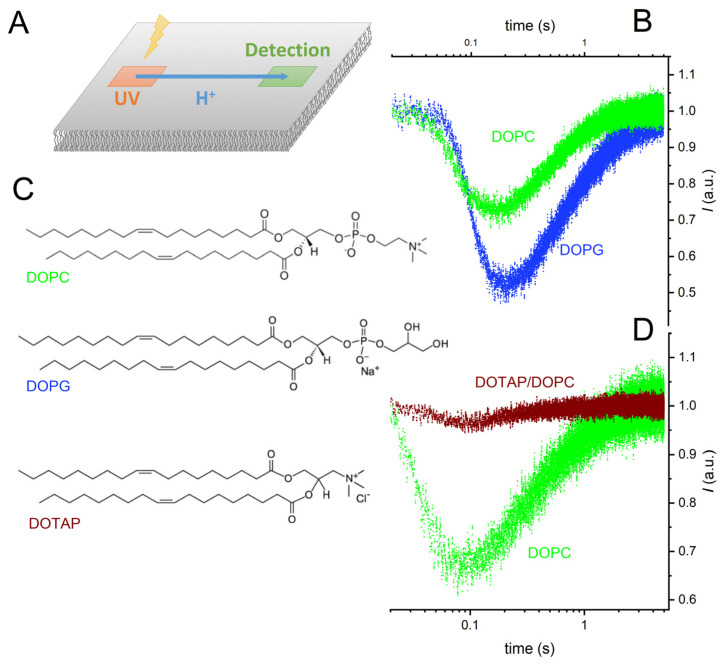
Representative records of interfacial proton migration. (**A**) Experimental scheme. Exposure of a 10 × 10 µm² area to a UV flash released protons from a caged compound that was adsorbed to the surface of a free-standing planar bilayer. The fluorescence of a membrane-anchored pH sensor in a distant 10 × 10 µm²-sized membrane patch changed upon proton arrival. (**B**) The changes in fluorescence intensity measured at a distance of 80 µm with negatively charged bilayers (1,2-dioleoyl-sn-glycero-3-phospho-(1′-rac-glycerol), DOPG) and uncharged bilayers (1,2-dioleoyl-sn-glycero-3-phosphocholine, DOPC) differ from each other. (**C**) Structural formulas of the three lipids: DOPC, DOPG, and 1,2-dioleoyl-3-trimethylammonium-propane (DOTAP). (**D**) The changes in fluorescence intensity at a distance of 50 μm are smaller for positively (DOTAP/DOPC mixture) charged membranes than for DOPC membranes. The four colored traces are data from representative individual release events. The buffer (pH = 9) contained 10 mM KCl.

**Figure 2 biomolecules-13-00352-f002:**
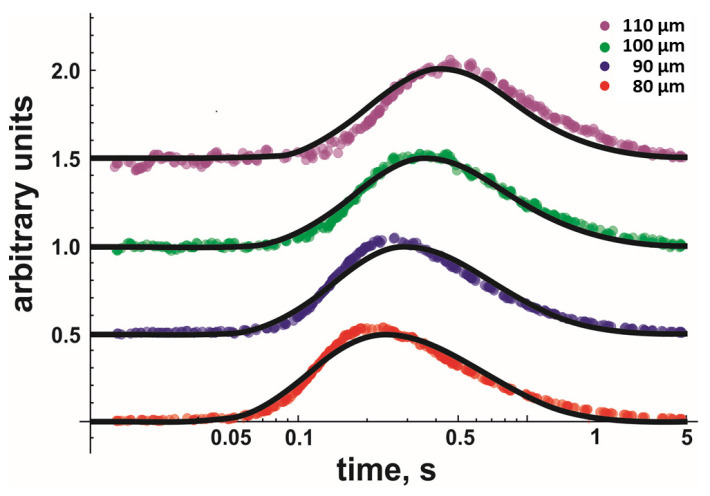
Kinetics of proton concentration changes adjacent to the DOPG membrane surface measured at different distances *x* from the release spot. The temperature was equal to 19 °C. The representative colored traces represent an average of 20 individual release events. The global fits of the non-equilibrium model (Equation (8)) to all traces measured for the four distances at 19 °C (also to those traces not shown here) are depicted as solid black lines. Data amplitude was normalized to 0.5, and the individual curves were shifted vertically for representational reasons.

**Figure 3 biomolecules-13-00352-f003:**
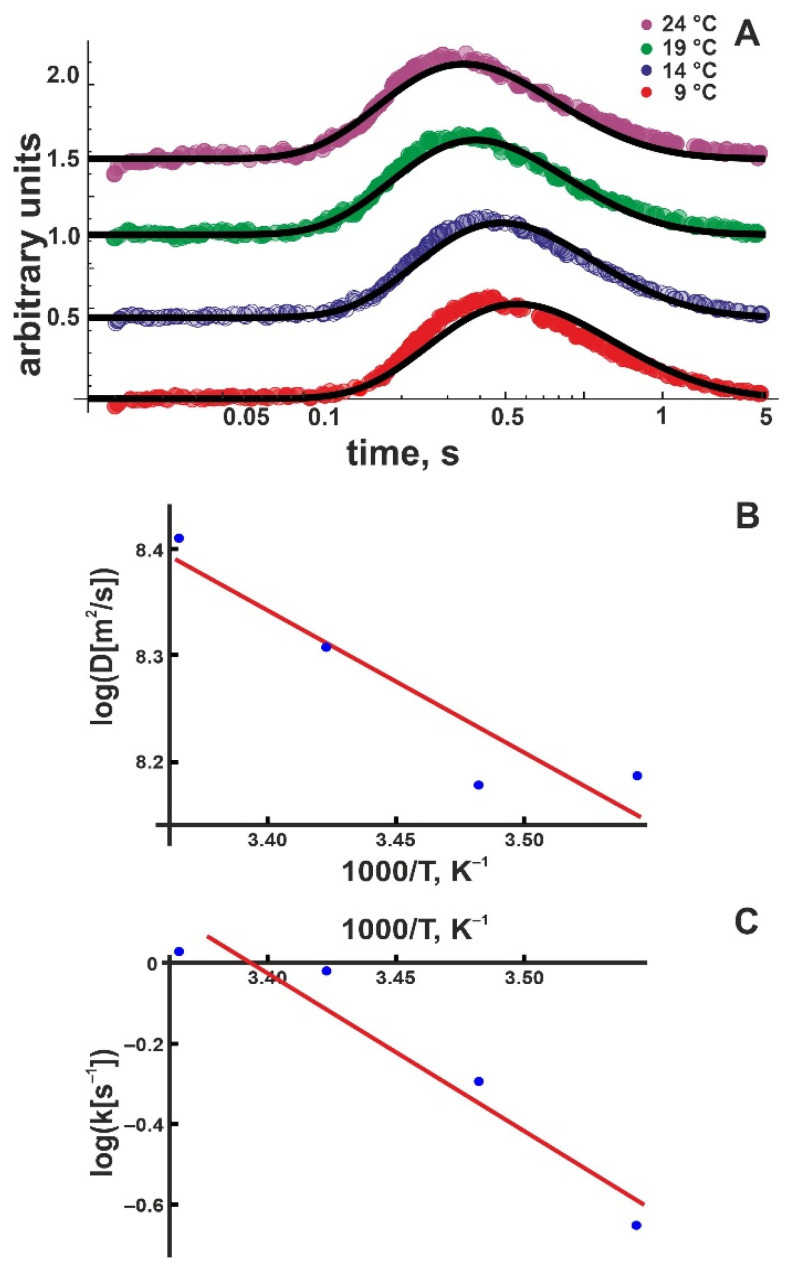
(**A**) Kinetics of the proton concentration adjacent to the DOPG membrane surface at 100 μm from the release spot for different temperatures. The colored traces are data from representative individual release events. The global fits of the non-equilibrium model (Equation (8)) to all traces at a given temperature (compare Figure 2) are depicted as solid black lines. Data amplitude was normalized to 0.5, and the individual curves were shifted vertically for better visibility. (**B**) Temperature dependence of the lateral diffusion coefficient *D* (in units μm^2^/s). The slope of the linear fit (red line) corresponds to $\Delta H_{D}^{‡}$ = (4.5 ± 1.2) *k*_B_*T*. We found the pre-exponential factor *A*_k_ = 6 × 10^5^ s^–1^. (**C**) Temperature dependence of the proton surface to bulk release constant, *k*_off_ (in units of s^−1^). The slope of the linear fit (red line) corresponds to $\Delta H_{k}^{‡}$ = (13 ± 3) *k*_B_*T*, the intercept to *A*_D_ = 1.3 × 10^7^ μm/s^2^.

**Figure 4 biomolecules-13-00352-f004:**
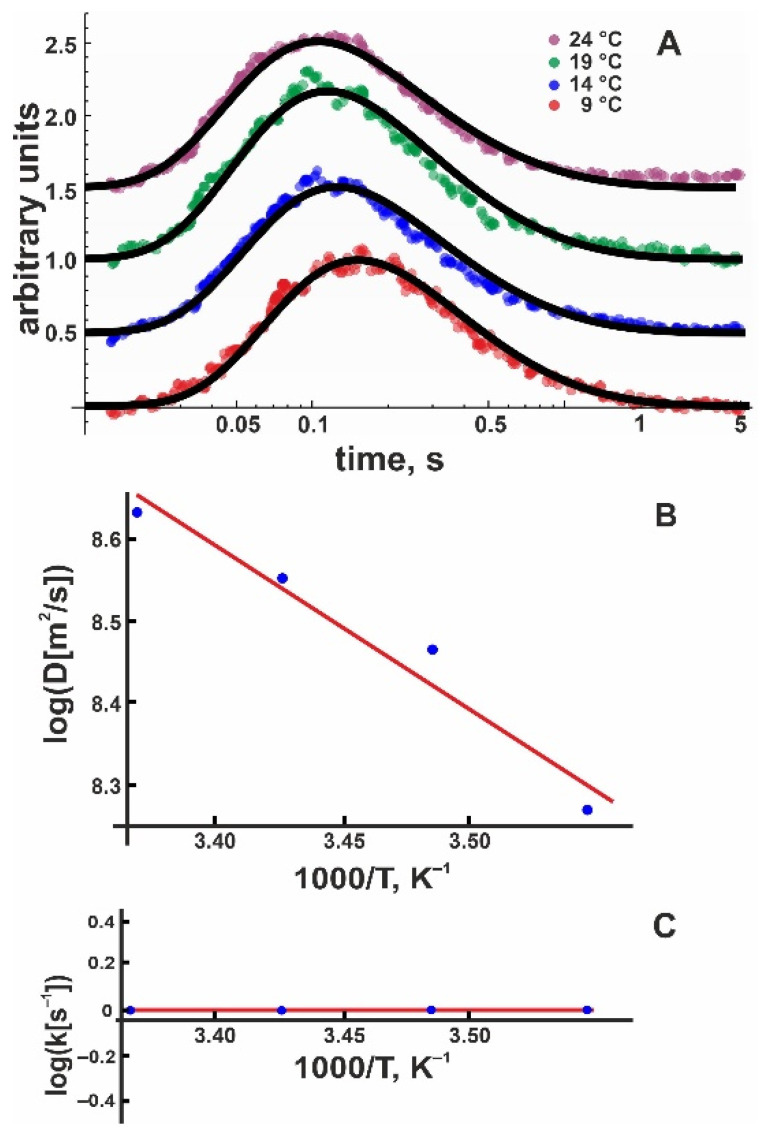
(**A**) Representative time traces of normalized proton concentration changes adjacent to the DOTAP/DOPC membrane surface at the indicated temperatures (colored traces). The distance from the release spot was equal to 50 μm. The solid black lines represent global fits of the non-equilibrium model (Equation (8)) to all traces at a given temperature. The individual curves were shifted vertically to avoid overlap. (**B**) Temperature dependence of the lateral diffusion coefficient (*D*, in units μm^2^/s). The slope of the temperature dependence corresponds to $\Delta H_{D}^{‡}$ = 6.6 ± 0.9 *k*_B_*T*. (**C**) The global fit of Equation (8) to the time traces (compare panel A) worked with *k*_off_ set to (1.2 ± 1.2) s^−1^ for all measured temperatures, indicating $\Delta H_{k}^{‡}$ = 0 ± 3 *k*_B_*T*.

**Table 1 biomolecules-13-00352-t001:** Measured zeta potential, ζ, of lipid vesicles of the indicated composition between 20 and 30 °C. The surface potential, $\psi_{0}$, was calculated assuming that the shear plane was located at a distance of 0.4 nm since the salt concentration was equal to 10 mM. We used Equation (13) to calculate the electrostatic surface–proton interaction energy, *E*.

	ζ (mV)	$\psi_{0}\;$	*E* (*k*_B_*T*)
DOPG	−88.3 ± 2.4	−110.2 ± 4.3	4.3 ± 0.2
DOPC	−2.4 ± 0.6	−2.7 ± 0.7	0.11 ± 0.02
DOTAP	74.6 ± 4.1	90.5 ± 5.8	−3.5 ± 0.2

**Table 2 biomolecules-13-00352-t002:** Summary diffusion coefficients and decay rates for proton kinetics near the DOPG and DOTAP/DOPC membranes (this work) and the DOPC membrane [21].

*T, C*	9	14	19	24	$\Delta H_{D}^{‡}$ (*k*_B_*T*)	$\Delta H_{k}^{‡}$ (*k*_B_*T*)	$\Delta H_{r}^{‡}$ (*k*_B_*T*)	$-T\Delta S_{r}^{‡}$ (*k*_B_*T*)
DOPC [21]								
*D* (μm^2^/s)	4584	4943	5126	6115	5.9 ± 1.1			
*k*_off_ (1/s)	1.9	2.1	2.3	2.7		5.7 ± 1.1	5.7 ± 1.1	26.1
DOPG								
*D* (μm^2^/s)	3591	3558	4059	4496	4.5 ± 1.2			
*k*_off_ (1/s)	0.52	0.75	0.98	1.04		13 ± 3	8.8 ± 3.2	20 ± 3.0
DOTAP + DOPC								
*D* (μm^2^/s)	3911	4753	5178	5616	6.6 ± 0.9			
*k*_off_ (1/s)	1					0 ± 3	3.5 ± 3.2	30 ± 3.0

## Data Availability

All data supporting reported results are presented in the Figures and Tables. Requests for original data should be directed to P.P.

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
