# Peer review of "Proton Migration on Top of Charged Membranes"

_biomolecules, 2023, doi:10.3390/biom13020352_

Round 1

Reviewer 1 Report

Weichselbaum et al describe a nice kinetic (out of equilibrium) analysis of migration of H+ on the surface of phospholipidic membranes. Based on a well designed experimental setup, the authors determine the time course of a fluorescent signal giving account of local H+ concentration at different temperatures. Then, the entlaphic and entropic components of the activation free energy barriers were determined for lateral H+ diffusion and H+ release to the bulk aqueous phase. Data analysis and conclusions are well supported by theory and experimental data. However, there are some minor points that need to be addressed in a revised version of the manuscript.

Suggestions:

1) The transition state theory (TST) analysis could be introduced earlier in the manuscript, following the Arrhenius like equations 6 and 7. This would permit to obtain accurate values for the enthalpic and entropic components of the activation free energies in a straightforward way, by fitting the TST models to the experimental data.

2) After the step-by-step analysis carried out in this work, a global fitting of all the model equations (eq 5, 6, 7 and TST) to the full set of experimental data at all the assayed temperatures for each phospholipidic membrane would allow a better estimation of all the parameter values with the corresponding standard errors.

Minor points

3) Page 3 lines 111 to 113. This paragraph seems to be a comment from the instruction to authors.

4) Page 4 lines 145-146. Please include a brief description of how the fluorescent intensity is converted into pH values.

5) Page 6 Table I. Please include the standard error of each value in the table.

6) Page 8 lines 271-274. This paragraph seems to be a comment from the instruction to authors

7) Page 11 reference 26. The source name is missing. Also check the abbreviations of the Journal names in other references.

Reviewer 2 Report

The text is very hard to read. The authors considered that the reader is familiar with their previous work, which was reflected in the experimental part. For a better reception by a wide range of readers, a more detailed, intended for the uninformed reader, description of the purpose of the research, the adopted model of interactions by the tested surfaces and the experimental methods used are necessary. Also, the role of proton transfer in the studied interactions should be preliminarily discussed before proton transfer is used as the main mechanism. Therefore, for a better understanding of the text, it is necessary to re-edition it so that it is legible for a wide audience.

Round 2

Reviewer 2 Report

The new, much expanded text of the publication is, in my opinion, very well written. Since the authors added the fragments that I mentioned in the review, the publication can be accepted for publication in its current form.

Author Response

Thanks.